# Multi-state Protein Sequence Design with DynamicMPNN

**Alex Abrudan,**[*]
Yusuf Hamied Department of Chemistry
University of Cambridge
Cambridge, UK
{aca41}@cam.ac.uk

**Sebastian Pujalte Ojeda,**[*]
Yusuf Hamied Department of Chemistry
University of Cambridge
Cambridge, UK
{sp2120}@cam.ac.uk

**Chaitanya K. Joshi**
Department of Computer Science and Technology
University of Cambridge
Cambridge, UK

**Matthew Greenig,**
Yusuf Hamied Department of Chemistry
University of Cambridge
Cambridge, UK

**Felipe Engelberger**
Institute for Drug Discovery
Leipzig University
Leipzig, Germany

**Alena Khmelinskaia**
Department of Chemistry
Ludwig-Maximilans-University
Munich, Germany

**Jens Meiler**
Institute for Drug Discovery
Leipzig University
Leipzig, Germany

**Michele Vendruscolo**
Yusuf Hamied Department of Chemistry
University of Cambridge
Cambridge, UK

**Tuomas P. J. Knowles**
Yusuf Hamied Department of Chemistry
University of Cambridge
Cambridge, UK

## Abstract

Structural biology has long been dominated by the *one sequence, one structure, one function* paradigm, yet many critical biological processes—from enzyme catalysis to membrane transport—depend on proteins that adopt multiple conformational states. Existing multi-state design approaches rely on post-hoc aggregation of single-state predictions, achieving poor experimental success rates compared to single-state design. We introduce DynamicMPNN, an inverse folding model explicitly trained to generate sequences compatible with multiple conformations through joint learning across conformational ensembles. Trained on 46,033 conformational pairs covering 75% of CATH superfamilies and evaluated using Alphafold 3, DynamicMPNN outperforms ProteinMPNN by up to 31% on decoy-normalized RMSD and by 12% on sequence recovery across our challenging multi-state protein benchmark.

## 1 Introduction

A commonly derived assumption from Anfinsen's experiment is that proteins adopt only one native 3D structure, leading to the "one sequence, one structure, one function" canon. This view has been indirectly reinforced by the predominant use of X-Ray crystallography in experimental protein structure determination, which requires that proteins form a homogenous, diffractible crystal to be characterised (Dishman & Volkman, 2018). The large collection of static protein structures in the PDB has enabled the development of high-accuracy machine learning models for tasks such as structure prediction (Jumper et al., 2021; Kryshtafovych et al., 2019) and inverse folding (Dauparas et al., 2022; Hsu et al., 2022). Amongst contemporary inverse folding models, ProteinMPNN has been particularly widely adopted in applied protein design projects due to its low inference costs and

---

[*]Equal contribution.

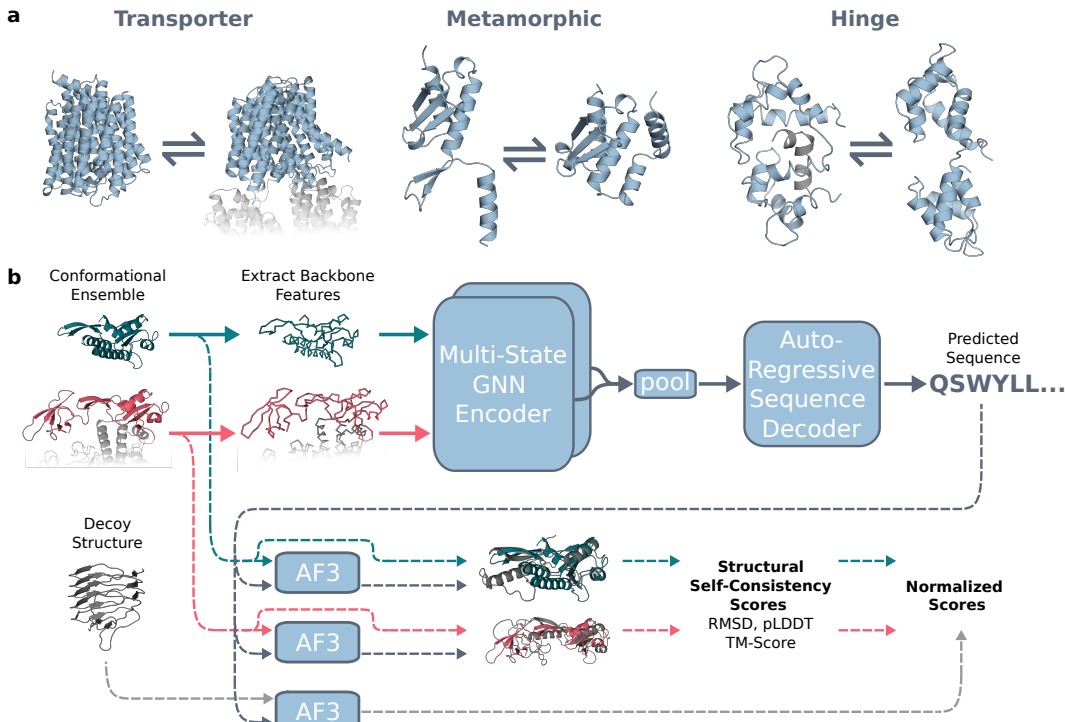

Figure 1: DynamicMPNN for multi-state protein design. (a) Examples of proteins with multiple conformational states: transporters in closed and open states (PDB: 6NC7, 6NC9), metamorphic protein with alternative folds (PDB: 4QHH, 4QHF) and hinges showing domain movement (PDB: 5D0W, 1CFC). (b) Schematic of DynamicMPNN, an inverse folding model trained to generate protein sequences with multiple conformational states. Conformations are encoded with their respective chemical environments (i.e. interaction partners shown in gray). Solid lines show the flow of information in the model, while dashed lines show the evaluation pipeline using AlphaFold 3 (AF3); employing target structures as templates during inference and measuring the deviations between predicted and target structures, with decoy structures serving as negative controls.

robust experimental success rates (Dauparas et al., 2022; Watson et al., 2023; Goverde et al., 2024), outperforming traditional physics-based design methods on both fronts (Liu & Kuhlman, 2006).

Although direct experimental characterisation of protein dynamics remains a challenge, the conformational diversity of proteins underlies crucial biological functions such as enzyme catalysis, protein-protein interactions, allostery, and human disease (Monzon et al., 2016). In applied protein design, bio-switches - proteins that switch between two structural states - are of particular importance, with key applications in engineering artificial bio-motors, signalling pathways, biosensors, or drug delivery systems (Stein & Alexandrov, 2015; Praetorius et al., 2023). While most known switches undergo rearrangements in the context of a single fold (Ambroggio & Kuhlman, 2006a), the class of metamorphic proteins undergo changes in both their secondary structure and fold (Fig. 1a) and have been predicted to represent up to 4% of the PDB chains (Porter & Looger, 2018). These proteins typically adopt two main functional states (Dishman & Volkman, 2018) and a finite number of conformations (see Discussion). Beyond the world of switches, other dynamic proteins are characterised by continuous conformational landscapes (e.g. intrinsically disordered proteins (Tompa & Fuxreiter, 2008)).

Multi-state protein design was first achieved through rational design and physics-based methods such as RosettaDesign (Liu & Kuhlman, 2006; Vucinic et al., 2020; Karimi & Shen, 2018). Previous campaigns leveraging these methods have attempted to design metamorphic metal-binding peptides (Ambroggio & Kuhlman, 2006b; Cerasoli et al., 2005), closely related sequences that adopt diverging folds (Wei et al., 2020), and hinge proteins with binder-regulated thermodynamic equilibria,

allowing the relative populations of different structural states to be modulated by exogenous proteins (Zhang et al., 2022; Quijano-Rubio et al., 2021). More recently, Praetorius et al. (2023) leveraged ProteinMPNN Multi-state Design (ProteinMPNN-MSD) (Dauparas et al., 2022) to link independent backbone states via step-wise logit averaging with shared autoregressive context. Multi-state ESM-IF (Hsu et al., 2022) employs an analogous strategy, differing primarily in aggregating probabilities (geometric mean) rather than logits (see Appendix A.3). ProteinGenerator (Lisanza et al., 2024) extends this principle to sequence diffusion, averaging logits across distinct structural conditioning inputs at each denoising step.

Despite these advances, current multi-state design pipelines have shown limited success. In Hsu et al. (2022), sequences designed with the dual-state strategy only showed marginally lower perplexity compared to sequences inverse-folded from one of the states and refoldability was not explored. Praetorius et al. (2023) used ProteinMPNN-MSD to design a de novo hinge protein with one sequence binding to a peptide: From an initial pool of over 2M computational designs conditioned on 28K similar de novo backbone pairs, a rigorous set of computational filters yielded only 9K sequences as likely candidates; experimental validation of a few sampled candidates showed a high success rate (Appendix A.3). Likewise, the authors of ProteinGenerator Lisanza et al. (2024) reported a significantly lower *in silico* design success rate of 0.05% for their multi-state backbone design task compared to rates of 2-10% observed for various single-state sequence design objectives using their framework (Lisanza et al., 2024). These observations - combined with the relative scarcity of published data on ML-driven multi-state *de novo* design campaigns - suggest that current ML methods for multi-state protein design have been significantly less successful than their single-state counterparts. We propose that this gap can be attributed to limited multi-conformational datasets, weak benchmarks, and the poor performance of folding models in predicting alternative states (Chakravarty et al., 2024), which adversely affects their efficacy as self-consistency filters in protein design workflows.

**Our contributions.** This paper introduces DynamicMPNN (Fig. 1b), a novel geometric deep learning-based pipeline for multi-state protein sequence design.

- DynamicMPNN is the first explicit multi-state inverse folding model for protein design. To train DynamicMPNN, we create a new ML-ready dataset of proteins with multiple conformations using the PDB and CoDNaS (Monzon et al., 2016) databases, and evaluate the method on 96 biologically relevant metamorphic, hinge, and transporter proteins.
- We introduce a novel data processing pipeline and architecture able to handle sequence-aligned structural ensembles with heterogeneous sequences, enabling us to leverage the conformational diversity across proteins with high sequence similarity.
- We propose a multi-state self-consistency metric and benchmark based on *Alphafold3* (AF3) (Abramson et al., 2024b) using target structures as templates.
- DynamicMPNN improves performance of ProteinMPNN (Dauparas et al., 2022) by up to 12% on sequence recovery, as well as 31% and 8% on decoy-normalized self-consistency values of the RMSD and TM-score, respectively.

## 2 THE DYNAMICMPNN PIPELINE

### 2.1 PROTEIN MULTI-CONFORMATIONAL DATASET

While over 900K individual protein chains (sequence-structure pairs) are available in the PDB, multi-conformational data is far more scarce with only roughly 12,000 NMR-derived protein ensembles covering just 21% of CATH superfamilies. To overcome this limitation, we exploit the sequence redundancy across the PDB to create build two multi-conformational datasets:

- **CoDNaS** We use the CoDNaS dataset (Monzon et al., 2016), which clusters PDB chains at ≥95% sequence similarity with unique UniProt IDs per cluster to prevent homologue leakage, yielding 46,033 clusters with varying numbers of conformations and covering 75% of CATH superfamilies (Fig. 3).

- **PDB80** To capture greater conformational diversity within training ensembles, we also employ ≥80% sequence similarity clusters available from the Protein Data Bank (Berman et al., 2000), yielding 46,924 clusters with at least two conformations (Fig. 2).

**Training-time data sampling.** From CoDNaS, we select the maximum-RMSD chain pair from each cluster to maximize conformational signal and reduce alignment artifacts[1] (Fig. 2a). For PDB80, pairs are sampled with probability proportional to their structural dissimilarity (1 - TM-score), biasing training toward larger conformational changes (Appendix A.4). We also explore extending the latter approach to 3 and 5 states (Section 2.2.2), though most known multi-state proteins switch between just two functional conformations (Dishman & Volkman, 2018; Leaver-Fay et al., 2011; Alberstein et al., 2022), and validating designs beyond two states remains experimentally challenging (Niazi, 2025). While our architecture supports arbitrary numbers of states, for the aforementioned reasons, we focus primarily on two-state design. Molecular dynamics datasets were not used due to limited sequence variability and scarcity of trajectories capturing large conformational changes (Vander Meersche et al., 2023; Mirarchi et al., 2024); disordered protein simulations (Tesei et al., 2024) were similarly excluded given their continuous conformational landscapes and low structural signal-to-noise ratio compared to globular multi-state proteins (Tompa & Fuxreiter, 2008).

**Dataset splitting.** We curate a benchmark composed of four previous studies of proteins with large 2-state conformational changes: (1) 92 metamorphic proteins (Porter & Looger, 2018), (2) 91 apo-holo proteins (Saldaño et al., 2022), (3) the OC23 and OC85 open-closed datasets (Kalakoti & Wallner, 2025), and (4) 20 transporter proteins (Kalakoti & Wallner, 2025). The proteins with the highest inter-state RMSD were assigned to the test set (96 samples), while the rest were assigned to the validation set (100 samples). Training clusters were filtered to exclude any with TM-score $> 0.4$ (Zhang & Skolnick, 2004; Xu & Zhang, 2010) and $> 30\%$ sequence similarity to test/validation structures, preventing structural similarity leakage and yielding a final training set of 44,243 conformer pairs. To evaluate designs beyond two states, we additionally curate a set of six proteins with well-characterized intermediate or flexible conformations: MBP (Wang et al., 2012), $\alpha$-hemolysin (Chatterjee et al., 2025), and Selecase (López-Pelegrín et al., 2013) (stable intermediates; 3-states), and Calmodulin, $\alpha$-synuclein (Chen et al., 2021), and A$\beta$-42 (flexible/disordered; 5-states).

We additionally curate a set of single-state sequence-structure pairs from the 30% sequence similarity clusters not represented in our multi-conformational training dataset ($n = 27,394$). This augmentation strategy maximizes coverage of protein fold space while preserving the multi-conformational learning signal. See Appendix A.4 for further details on dataset composition.

## 2.2 DYNAMICMPNN FOR MULTI-STATE INVERSE FOLDING

Single-state inverse folding methods seek to model the conditional distribution $p(Y|X)$ where $X \in \mathbb{R}^{n \times 3 \times 3}$ represents a protein backbone with $n$ residues, and $Y = (y_1, \ldots, y_n)$ is the amino acid sequence. Extensions of these methods to multi-state design have thus far been limited to post-hoc/decoder-level aggregation (A.3) of independent single-state predictions. We believe such methods favour logits highly biased towards one conformation, whose average over the states is higher than the one of moderately valued logits across both states (Joshi et al., 2025). Instead, DynamicMPNN learns the joint conditional distribution of $p(Y|X_1, \ldots, X_m)$ directly through autoregressive sequence generation, where $\{X_1, \ldots, X_m\}$ represent distinct protein conformations encoded into a shared latent space; thus learning a sequence distribution that simultaneously satisfies multiple structural constraints. We decompose this joint conditional probability using the autoregressive factorization:

$$p(Y|X_1, ..., X_m) = \prod_{i=1}^{n} p(y_i | y_{i-1}, ..., y_1; X_1, ..., X_m) \tag{1}$$

where each factor represents the probability of selecting residue $y_i$ given the sequence prefix and the complete structural ensemble.

---

[1]Note that aligning more sequences introduces additional gap tokens (i.e. reducing sequence-structure overlap) diluting the conformational signal.

**Overall architecture**. DynamicMPNN independently encodes each of the functional states of a protein, together with their binding partners, into a shared latent feature space (Fig. 1b). Embeddings of the chains-to-be-designed are then pooled across conformations to obtain a single embedding from which a sequence is auto-regressively generated.

Our architecture builds upon gRNAde (Joshi et al., 2025), a multi-state GNN model for RNA inverse folding. For both encoder and decoder, we employ SE(3)-equivariant Geometric Vector Perceptron (Jing et al., 2021) layers which maintain computational efficiency through edge sparsity (k-NN edges with k=32). Within the GVP, scalar features $s_i \in \mathbb{R}^{k \times f}$ and vector features $\vec{v}_i \in \mathbb{R}^{k \times f' \times 3}$ are defined for each node $i$ (Duval et al., 2024):

$$\boldsymbol{m}_i, \vec{\boldsymbol{m}}_i \coloneqq \sum_{j \in \mathcal{N}_i} \mathrm{MSG}\big( (\boldsymbol{s}_i, \vec{\boldsymbol{v}}_i), (\boldsymbol{s}_j, \vec{\boldsymbol{v}}_j), \boldsymbol{e}_{ij} \big) \tag{2}$$

$$\boldsymbol{s}_i', \vec{\boldsymbol{v}}_i' \coloneqq \mathrm{UPD}\big( (\boldsymbol{s}_i, \vec{\boldsymbol{v}}_i) , (\boldsymbol{m}_i, \vec{\boldsymbol{m}}_i) \big) \tag{3}$$

where MSG, UPD are Geometric Vector Perceptrons - a generalization of Multi Layer Perceptrons that takes as input and updates scalar and vector features along separate channels in order to achieve $O(3)$-equivariant message passing. The overall GNN encoder is $SO(3)$-equivariant due to the use of reflection-sensitive input features (dihedral angles) combined with $O(3)$-equivariant GVP-GNN layers Joshi et al. (2025). Both the encoder and decoder are assigned 8 GVP-GNN layers, following findings in Hsu et al. (2022) (2022). See Appendix A.5 for further details.

We present two different encoder architectures:

- **DynamicMPNN** uses independent encoder channels for each conformation, followed by Deep Set pooling (Zaheer et al., 2017) - it is invariant to conformation order and does not add extra parameters to the model. We note that while some more expressive pooling strategies have been shown to provide marginal performance improvements, they usually come at a great cost in efficiency (Joshi et al., 2025). Only node features are updated.

- **DynamicMPNN + DSS** implements cross-attention between the encoder channels after each layer using a Deep Symmetric Set (DSS) (Maron et al., 2020) module, which allows for richer inter-conformation interactions at the cost of increased computational cost. In the scatter/gather DSS strategy, node embeddings of all design chains are averaged, passed through GVP layers, and added back to features of each channel through a residual connection. Both node and edge features are updated.

**Heterogeneous sequence processing.** A key architectural contribution is our handling of non-identical sequences across conformations, necessary for exploiting the full conformational diversity in the PDB. Aligning non-identical sequences introduces gap tokens, and X-ray structures often contain unresolved residues; together resulting in ensembles of heterogeneous composition (i.e. varying lengths, missing residues, alignment gaps). We address this with the following protocol: (1) cluster chain members are sequence-aligned prior to featurization; (2) paired PDB complexes are featurized and encoded independently; (3) gap positions are masked and excluded from message passing; (4) during pooling, embeddings from all cluster members are extracted, stacked, and pooled, with gap-node embeddings zeroed out. This preserves all available structural information while incorporating the context of the surrounding chemical environment (i.e. binding partners). To prevent leakage, during training we mask sequence information for chains with $> 70\%$ similarity to the ground truth.

### 2.2.1 MULTI-CHAIN STRATEGY

Unlike previous work (Joshi et al., 2025), DynamicMPNN processes full PDB entries containing sampled cluster chains, enabling the model to condition conformational changes on binding partners and oligomeric states. This opens possibilities for engineering controllable conformational switches by tuning free energy differences between folds and their binding interactions (Alberstein et al., 2022). Note that while explicitly a 2-state approach, multi-chain training implicitly exposes the model to additional states when PDB entries contain multiple cluster conformations (Fig. 2d). As discussed in Section 2.1, most multi-state proteins switch between only two conformations (Dishman & Volkman, 2018; Leaver-Fay et al., 2011; Alberstein et al., 2022), making 2-state design the most biologically relevant regime.

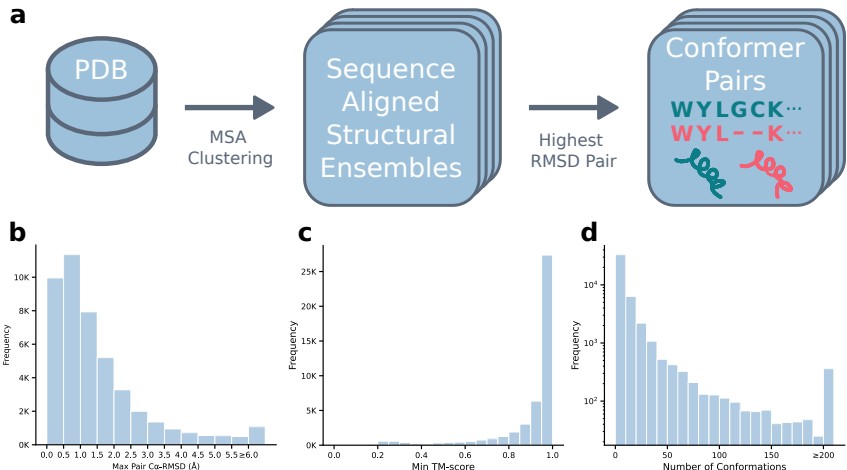

Figure 2: Multi-state protein dataset. (a) Data processing pipeline: structures from the PDB are clustered by sequence identity at a given threshold (e.g. 80%), to form sequence aligned structural ensembles and the highest-RMSD pair is selected from each cluster for training. (b–d) Dataset statistics for PDB clustered at 80% sequence identity (see Appendix A.4 for details). Distribution of the maximum C$\alpha$-RMSD (b) and minimum TM-score (c) between pairs of structures in each cluster. (d) Distribution of the number of conformations per cluster.

### 2.2.2 SINGLE-CHAIN STRATEGY

To efficiently run ablations for $k > 2$ states—training time and memory scale linearly with the number of encoded states—we implement a single-chain strategy encoding only cluster chains without binding partners. Single-chain training is performed on PDB80, which provides sufficient cluster diversity for sampling $k \in \{2, 3, 5\}$ states via TM-score weighted sampling. We evaluate these models on 6 proteins with well-characterized multi-state behavior (Tables 6, 7) and directly compare k=2 single-chain versus multi-chain performance (Table 3). Extending multi-chain training to $k > 2$ states is straightforward and left for future work.

### 2.3 MULTI-STATE DESIGN EVALUATION

Given the high degeneracy of the sequence-to-structure mapping, where divergent sequences can adopt identical folds (Rost, 1999; Sander & Schneider, 1991), sequence recovery is often an insufficient metric for design success. Therefore, following previous work Wang et al. (2023), we also evaluate the refoldability of generated sequences (Appendix A.2). Existing refoldability methods compare target structures to single conformations predicted by folding models such as AlphaFold2 (Jumper et al., 2021). We argue that this unconstrained approach is unsuitable for multi-state design since folding models typically predict one dominant state or an unphysical interpolation, failing to sample the full conformational ensemble Lane (2023); Chakravarty et al. (2024); Saldaño et al. (2022).

**Template-based Alphafold 3 refoldability.** To address these sampling limitations, we propose a template-based AlphaFold 3 (AF3) framework Abramson et al. (2024b) to explicitly verify structural compatibility - we adapt the findings of Roney & Ovchinnikov (2022) to our multi-state setting (see Appendix A.6). While unconstrained DL models may fail to spontaneously find alternative states, we leverage AF3's template mechanism to assess the compatibility between the designed sequence and a true template. For each de novo sequence, we perform 2 AF3 runs, one with each conformational state provided as template, shifting the evaluation from structural prediction to compatibility assessment. The similarity - C$\alpha$-RMSD or TM-score (Zhang & Skolnick, 2004) - between predicted structures and the ground-truth templates, along with AF3 confidence scores, serve as a proxy for the likelihood that the designed sequence can adopt the target states.

Formally, for a protein with conformational states $X = \{X_1, X_2, \ldots, X_m\}$ and designed sequence $Y$, we define the AF3 RMSD for each target conformation $X_k$ as:

$$\text{AF3}_{template}(Y, X_k) = \text{RMSD}(\text{AF3}(Y, X_k), X_k) \tag{4}$$

where $\text{AF3}_{template}(Y, X_k)$ is the structure predicted by AlphaFold3 for sequence $Y$ when a template of $X_k$ is provided. To account for the structural bias induced by the template, we define a normalization strategy to contextualize observed deviations:

**Decoy normalization (Decoy Norm)**: We provide AF3 structurally dissimilar decoy structures as templates (TM-score $< 0.4$) using the same sequences designed and measure the resulting deviations. This control assesses whether sequences fold specifically into their targets or are compatible with arbitrary structures:

$$\text{RMSD}_{\text{decoy}}(Y, X_k; D) = \frac{\text{AF3}_{template}(Y, X_k)}{\text{AF3}_{template}(Y, D)} \tag{5}$$

where $D$ is a decoy that is structurally dissimilar to $X_k$. As noted in Roney & Ovchinnikov (2022), when provided a false decoy, AF2 globally aligns its prediction with the decoy, but it senses the local inconsistencies between the sequence and decoy backbone structure - AF2 therefore relaxes the structural prediction locally with respect to the decoy and correctly yields low self-consistency metrics. Additionally, we measure pLDDT confidence scores to evaluate AF3 fold uncertainty. High RMSD with low pLDDT indicates poor template matching, while low RMSD with high pLDDT suggests a successful design. The same decoy normalization strategy is also applied to TM-score and pLDDT metrics.

**Template-free baseline.** To quantify the impact of template conditioning, we additionally evaluate refoldability without structural guidance by omitting the template input in AF3 (i.e. standard AF3 pipeline). Comparing these predictions against both target conformations reveals: (1) whether templates overly bias predictions toward target conformations, and (2) whether AF3 can recover alternative states or collapses to a single dominant structure.

**Generative ensemble evaluation.** As an orthogonal evaluation approach, we employ BioEmu (Lewis et al., 2025), a generative model trained to sample from protein conformational equilibrium distributions. BioEmu represents a class of emerging generative models specifically trained to approximate conformational ensembles. We selected BioEmu over alternative ensemble generators such as AlphaFlow (Jing et al., 2024) or PepFlow (Abdin & Kim, 2023) as it represents the current state-of-the-art (Jing et al., 2025), trained on orders of magnitude more data than competing methods, including molecular dynamics equilibrium distributions and conformational clusters from AFDB (Varadi et al., 2024). However, current ensemble generation methods like BioEmu and AlphaFlow are restricted to single protein chains, precluding evaluation of refoldability in the presence of binding partners that often drive conformational changes and likely underestimating our designs' true multi-state capacity. For each designed sequence, we sample 50 structures (Fig. 5) and measure the maximum TM-score and minimum RMSD achieved against each target conformation.

Table 1: Refoldability performance comparison of DynamicMPNN model variants on multi-state protein design benchmark ($n = 96$). Raw metrics show absolute performance values, while normalized metrics show performance relative to random decoy structures.

| Model Variant | Raw Metrics | | | Decoy-Normalized Metrics | | |
|---|---|---|---|---|---|---|
| | pLDDT ↑ | RMSD (Å) ↓ | TM-score ↑ | pLDDT ↑ | RMSD ↓ | TM-score ↑ |
| Combined Training | 82.08 (7.62) | 2.35 (2.51) | 0.870 (0.162) | 1.354 (0.410) | **0.124** (0.130) | **6.684** (1.485) |
| Combined Training + DSS | 81.61 (7.23) | 2.56 (2.36) | 0.862 (0.158) | 1.398 (0.442) | 0.131 (0.125) | 6.627 (1.514) |
| Single Training | 68.35 (15.46) | 8.16 (9.96) | 0.652 (0.322) | 1.383 (0.467) | 0.348 (0.336) | 4.830 (2.285) |
| Sampled Pair Training | **82.26** (7.61) | **2.29** (1.92) | **0.872** (0.161) | **1.470** (0.466) | 0.125 (0.129) | 6.630 (1.506) |
| Sampled Pair Training + DSS | 81.88 (7.85) | 2.45 (2.33) | 0.865 (0.158) | 1.436 (0.470) | 0.127 (0.121) | 6.668 (1.466) |

## 3 RESULTS AND DISCUSSION

**Setup.** We evaluate how exposure to multi-state training data affects design performance. Our primary comparison is between *Single Training* (single-state pairs only, analogous to standard inverse-folding models) and *Combined Training* (mixing single-state and multi-state samples). We additionally explore multi-staged training where single-state pretraining precedes multi-state finetuning (*Multi-Finetuning*.

Table 2: Refoldability performance comparison of DynamicMPNN model variants on template-less multi-state protein design benchmark ($n = 96$).

| | Template-less both states | | | Template-less worst state | | | With Template worst state | | |
|---|---|---|---|---|---|---|---|---|---|
| Model Variant | pLDDT ↑ | RMSD (Å) ↓ | TM-score ↑ | pLDDT ↑ | RMSD ↓ | TM-score ↑ | pLDDT ↑ | RMSD ↓ | TM-score ↑ |
| Combined Training | 81.87 (7.99) | 4.38 (4.05) | 0.78 (0.20) | 79.72 (9.89) | 6.18 (5.92) | 0.69 (0.27) | 80.43 (9.50) | 3.05 (3.41) | 0.82 (0.22) |
| Single Training | 70.11 (12.41) | 10.87 (10.48) | 0.54 (0.31) | 67.65 (13.67) | 12.56 (11.30) | 0.47 (0.32) | 66.49 (16.09) | 9.34 (10.52) | 0.59 (0.35) |

Table 3: Sequence recovery performance comparison across DynamicMPNN model variants and ProteinMPNN baseline on multi-state protein design benchmark.

| Model Variant | Sequence Recovery (%) ↑ |
|---|---|
| Combined Pretraining + Multi Finetuning | **42.7** (8.8) |
| Single Pretraining + Multi Finetuning | 42.1 (8.3) |
| Combined Pretraining + Multi Finetuning + DSS | 41.9 (8.6) |
| Combined Training | 41.0 (8.5) |
| Single Pretraining + Multi Finetuning + DSS | 40.3 (8.2) |
| Combined Training + DSS | 38.8 (7.8) |
| ProteinMPNN MSD* | 38.0 (11.0) |
| Single chain 2-state | 37.35 (9.04) |
| Single Training | 27.1 (9.4) |
| Single Training + DSS | 26.2 (8.5) |

Table 4: Performance comparison of DynamicMPNN variants and ProteinMPNN baseline on subset ($n = 61$) of multi-state protein design benchmark. Standard deviations shown in parentheses. Note that ProteinMPNN MSD's handling of gap tokens and missing residues (i.e., X tokens) limited the number of designs that could be refolded using AF3, necessitating separate comparison.

| | Raw Metrics | | | Decoy-Normalized Metrics | | |
|---|---|---|---|---|---|---|
| Model | pLDDT ↑ | RMSD (Å) ↓ | TM-score ↑ | pLDDT ↑ | RMSD ↓ | TM-score ↑ |
| Combined Training | 82.22 (6.76) | **2.27** (1.59) | 0.849 (0.166) | 1.266 (0.332) | **0.129** (0.109) | 6.482 (1.340) |
| Combined Training + DSS | 82.00 (6.66) | 2.65 (2.40) | 0.836 (0.176) | 1.286 (0.336) | 0.144 (0.126) | 6.513 (1.586) |
| Single Training | 69.35 (14.45) | 8.00 (9.44) | 0.623 (0.324) | 1.286 (0.396) | 0.376 (0.338) | 4.455 (2.186) |
| Sampled Pair Training | 81.89 (7.36) | 2.28 (1.75) | **0.850** (0.179) | **1.370** (0.405) | 0.134 (0.127) | 6.449 (1.521) |
| Sampled Pair Training + DSS | **82.45** (6.96) | 2.58 (2.47) | 0.839 (0.177) | 1.341 (0.431) | 0.139 (0.120) | **6.529** (1.570) |
| ProteinMPNN MSD* | 79.55 (9.75) | 3.31 (2.88) | 0.806 (0.207) | 1.326 (0.348) | 0.187 (0.191) | 6.054 (1.771) |

Table 5: BioEmu refoldability comparison of DynamicMPNN model variants on test set ($n = 96$). Standard deviations in parentheses.

| | Performance Metrics | | Success Rate (%) | | |
|---|---|---|---|---|---|
| Model Variant | TM-score ↑ | RMSD (Å) ↓ | $TMS_{0.7}$ ↑ | $TMS_{0.8}$ ↑ | $TMS_{0.9}$ ↑ |
| Combined Training | **0.623** (0.230) | **5.66** (4.82) | **37.5** | **22.9** | **11.5** |
| Single Training | 0.394 (0.252) | 13.28 (10.27) | 17.7 | 7.3 | 3.1 |

Table 6: Comparison between 2-state and 3-state single chain models with average pooling or DSS for 3 curated 3-state proteins.

| | Sequence Recovery (%) ↑ | | | | TM-score ↑ | | | |
|---|---|---|---|---|---|---|---|---|
| Protein | 2-state | 2-state + DSS | 3-state | 3-state + DSS | 2-state | 2-state + DSS | 3-state | 3-state + DSS |
| MPD | 50.3 (1.01) | 62.3 (1.03) | 51.3 (0.98) | 56.1 (1.07) | 0.93 (0.04) | 0.90 (0.04) | 0.88 (0.02) | 0.89 (0.04) |
| Selecase | 42.9 (2.81) | 51.3 (2.21) | 45.4 (1.22) | 45.0 (1.90) | 0.77 (0.03) | 0.81 (0.01) | 0.81 (0.03) | 0.81 (0.01) |
| $\alpha$-hemolysin | 32.3 (0.87) | 31.4 (1.24) | 33.0 (1.10) | 32.4 (0.92) | 0.93 (0.03) | 0.91 (0.03) | 0.94 (0.02) | 0.91 (0.03) |

We train on two multi-conformational datasets and explore several architectural variations:

Table 7: Comparison between 2-state and 5-state single chain models with average pooling or DSS for 3 curated highly flexible proteins.

| Protein | Sequence Recovery (%) ↑ | | | | TM-score ↑ | | | |
|---|---|---|---|---|---|---|---|---|
| | 2-state | 2-state + DSS | 5-state | 5-state + DSS | 2-state | 2-state + DSS | 5-state | 5-state + DSS |
| A$\beta$-42 | 27.8 (2.25) | 9.8 (1.21) | 17.4 (2.00) | 16.8 (1.74) | 0.08 (0.05) | 0.16 (0.01) | 0.08 (0.04) | 0.07 (0.05) |
| $\alpha$-synuclein | 15.7 (1.20) | 15.8 (1.47) | 17.6 (2.80) | 24.2 (1.71) | 0.07 (0.02) | 0.10 (0.03) | 0.15 (0.01) | 0.12 (0.08) |
| Calmodulin | 34.2 (1.10) | 34.0 (1.48) | 43.4 (0.80) | 43.0 (1.09) | 0.47 (0.15) | 0.46 (0.11) | 0.51 (0.05) | 0.58 (0.05) |

- **CoDNaS:** Multi-chain models encoding full PDB complexes including binding partners. Maximum RMSD conformer pairs are pre-selected per cluster prior to training.
- **PDB80 (Sampled Pair Training):** Multi-chain models with TM-score weighted sampling to dynamically select conformer pairs during training, capturing greater conformational diversity.
- **Single-chain ablations:** To efficiently ablate designs beyond two states, we train single-chain models (encoding only target chains without binding partners) on PDB80, dynamically sampling up to $k \in \{2, 3, 5\}$ conformational states per cluster using the same TM-score weighted scheme.

We also compare architectures with and without DSS cross-attention modules across these configurations. All models were trained on either 8 A100-80GB or 8 H100-80GB GPUs using a batch size of 32 and Adam (Kingma, 2014) optimizer with learning rate $10^{-3}$. Training for each stage was run until convergence of performance on the validation set, typically after 20-40 epochs ($\approx$17-34 hours).

Then, DynamicMPNN models and ProteinMPNN (using Multi-state Design inference strategy) were used to sample 16 sequences for each of the 96 benchmark test proteins (see Table 8 for DynamicMPNN inference costs and model size).

**Evaluation protocol.** For each designed sequence, we evaluate refoldability using the template-based pipeline described above, sampling 5 structures per sequence. For each target conformation, we select the best-performing sample (i.e., lowest RMSD, highest TM-score, highest pLDDT), then aggregate metrics by averaging across the 16 designed sequences. The same protocol is applied with decoy templates for normalization.

Additionally, for sequences designed using *Combined Training* and *Single Training* models, we conduct refoldability evaluations with BioEmu and template-free AF3. For BioEmu (Lewis et al., 2025), we sample 50 structures per sequence and report the best-performing sample per target conformation averaged across designs. We define success rates (TMS$_\tau$) as the percentage of targets for which at least one designed sequence has both conformational states recovered with maximum TM-score (across the sampled ensemble) exceeding threshold $\tau \in \{0.7, 0.8, 0.9\}$.

For template-free AF3, we apply the same evaluation protocol as template-based AF3. To assess whether designs consistently recover both states, we additionally report worst-state metrics—the performance on whichever conformational state is harder to recover for each design—for both template-based and template-free evaluations (Table 2).

**DynamicMPNN outperforms existing benchmarks across multiple evaluation metrics.** Our best-performing model achieves substantial improvements over baseline methods: a 31% reduction in decoy-normalized RMSD (Tab. 4) and a 12% improvement in sequence recovery (Fig.4; Tab. 3) compared to ProteinMPNN Multi-State Design (MSD). This performance gain is particularly noteworthy considering that ProteinMPNN's training dataset contains proteins within 30% sequence similarity clusters of 91 out of 96 benchmark proteins, making our improvement on this established baseline significant. We leave the retraining of ProteinMPNN on a rigorous train-test split for future work.

**Combined and multi-state finetuning training strategies far outperform the single-state only training.** A more rigorous comparison involves our single-state trained model, which adheres to the same training-test split and thus eliminates potential data leakage concerns. Models trained only on single-state data (Single Training) perform poorly on multi-state design as they are unable to create and decode a meaningful latent representation of the conformational changes. The Combined Training and approaches, which exposes models to multi-state examples during training, achieve the opti-

mal balance and consistently outperform Single Training as well as other models across all metrics. (Tab. 3, 1). Some likely reasons to explain the under performance of our Single Training checkpoint with respect to ProteinMPNN is the aforementioned unfair train splitting of ProteinMPNN as well as the multi-state latent space embeddings being out-of-distribution for the decoder parameters that have only been trained on single state data. Additionally, versions trained on PDB80 with TM-weighted sampling slightly outperform those trained on CoDNaS with MAX TM-score sampling, likely due to the increased diversity of conformational pairs seen during training 1.

**Fold-switch designs show best improvements over ProteinMPNN.** Analyzing performance stratified by the protein class in the test set (i.e. metamorphic, hinge, and transporter), (Fig. 6a, b), we observe that DynamicMPNN's advantage over ProteinMPNN is most pronounced in fold-switching proteins (metamorphic). This shows that DynamicMPNN can effectively fit the most complex conformational changes that undergo complex rearrangements of both tertiary and secondary structure (Dishman & Volkman, 2018). The distributions of AF3-predicted TM-scores over the inter-state TM-scores are plotted in Fig. 7.

**Template-free AF3 is not suitable for evaluating multi-state designs.** As has been previously shown for AlphaFold 2 (AF2) Chakravarty & Porter (2022), without structural guidance, AF3 collapses to a single dominant conformation, with worst-state TM-scores degrading from 0.82 (template-based) to 0.69 for Combined Training designs (Table 2). This inability to recover alternative states motivates our template-based evaluation framework: we expect AF3 to be reliable with respect to sequence-template folding compatibility estimation, based on Roney & Ovchinnikov (2022). Notably, Combined Training outperforms Single Training across all conditions—template-free average (0.78 vs 0.54), worst-state (0.69 vs 0.47), and template-based (0.82 vs 0.59)—confirming multi-state training benefits persist regardless of evaluation protocol.

**BioEmu mirrors AF3-template results.** Both AF3 and BioEmu evaluation frameworks demonstrate consistent model rankings and highly correlated refoldability scores (Pearson $r = 0.71$ for TM-score, $r = 0.79$ for RMSD; decoy-normalized $r = 0.58$, $r = 0.61$), with Combined Training consistently outperforming Single Training. This further emphasizes that regardless of the in silico evaluation metric that we is used, models trained with multi-state data consistently outperform those without.

**The optimal number of states is highly dependent on the protein system.** While 2-state modelling is the most biologically relevant approach for most multi-state proteins, a question worth exploring is whether DynamicMPNN can be generalised to proteins with metastable states, intermediates or even proteins with flatter energy landscapes. We compared the performance of the 2-state single chain model to the 3-state and 5-state respectively on 3 proteins each: (1) 3 proteins with 2 main states with a stable intermediate (MPD, $\alpha$-hemolysin, Selecase - Table 6); (2) 3 highly flexible proteins, either with disordered linkers (calmodulin) or fully disordered and which fold upon binding ($\alpha$-synuclein, A$\beta$-42) - Table 7. Results show how including extra states can be both beneficial (calmodulin) as well as detrimental (MPD), as the model has to learn to reconcile more diverse structural information. The inclusion of DSS is equally inconclusive. This is probably due to the lack of such proteins in the training set. To assess whether the single chain nature of these models impacts the metrics, we also compare the 2-state single chain to the 2-state multi chain model, observing a moderate albeit expected drop in performance (Table 3).

## 4 CONCLUSION

We present DynamicMPNN, the first explicit multi-state inverse folding model, achieving up to 31% improvement over ProteinMPNN on our multi-state benchmark. By jointly learning across conformational ensembles rather than aggregating single-state predictions, DynamicMPNN captures sequence constraints required for multiple functional conformations. Critically, models trained with multi-state data consistently outperform single-state trained models across all evaluation metrics (template guided AF3, AF3 template-free, and BioEmu) demonstrating robust benefits of explicit multi-state training regardless of evaluation framework. This opens possibilities for engineering synthetic bioswitches, allosteric regulators, and molecular machines. It is unclear if the presented one-size-fits-all approach to multi-state design will be effective experimentally, or if specialized models for different classes of conformational changes depending on their thermodynamic complexity will be beneficial.

## SOFTWARE AND DATA

We provide the code for our model in the following repository: `github.com/Alex-Abrudan/DynamicMPNN`.

## ACKNOWLEDGEMENTS

The authors would like to thank Rohith Krishna, Adam Broerman, Kai Yi, Simon Mathis, Vladimir Radenkovic, Michaela Brezinova, Gustavo Parisi, and Pietro Liò for insightful comments and discussions. SPO was supported by the Una and Derek Finlay scholarship. CKJ was supported by the A*STAR Singapore National Science Scholarship (PhD). AK was supported by the Graduate School of Quantitative Biosciences Munich (QBM). DynamicMPNN was trained using computational resources provided by the CSD3 Cambridge HPC cluster and Modal's GPU infrastructure.

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

## A APPENDIX

### A.1 LLM ACKNOWLEDGEMENT

The authors acknowledge that they have used LLMs in the process of paper writing for style suggestions and proofreading.

### A.2 SUPPLEMENTARY RESULTS

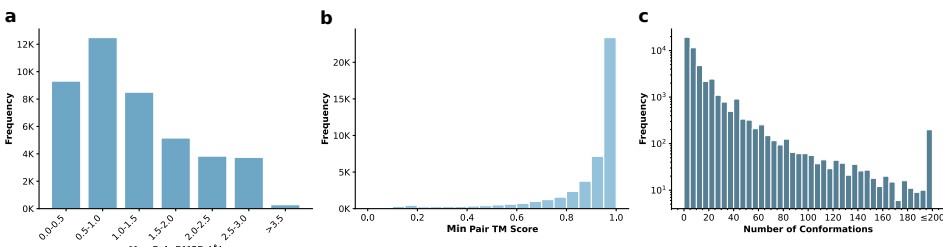

Figure 3: Overview of CoDNaS dataset. Distribution of the maximum C$\alpha$-RMSD (a) and minimum TM-score (b) between pairs of structures in each CoDNaS cluster. (c) Distribution of the number of conformations per CoDNaS cluster.

Table 8: Inference cost for sampling 16 sequences using one NVIDIA A100-80GB GPU and 32 AMD EPYC 7763 CPU cores. Residue count includes target and binding partner chains. DynamicMPPN model sizes: 4.86M parameters (+DSS) and 4.23M parameters (no DSS).

| | Time (s) | | Peak GPU Memory (MB) | |
|---|---|---|---|---|
| **Total Residues** | **+DSS** | **No DSS** | **+DSS** | **No DSS** |
| 250 | 47.7 | 28.6 | 1067.1 | 534.3 |
| 500 | 86.4 | 51.7 | 1568.6 | 788.2 |
| 1000 | 156.5 | 93.5 | 2305.8 | 1162.9 |
| 2000 | 283.3 | 169.1 | 3389.6 | 1715.6 |
| 4000 | 512.9 | 305.8 | 4982.7 | 2531.1 |
| 8000 | 928.5 | 552.8 | 7324.6 | 3734.3 |

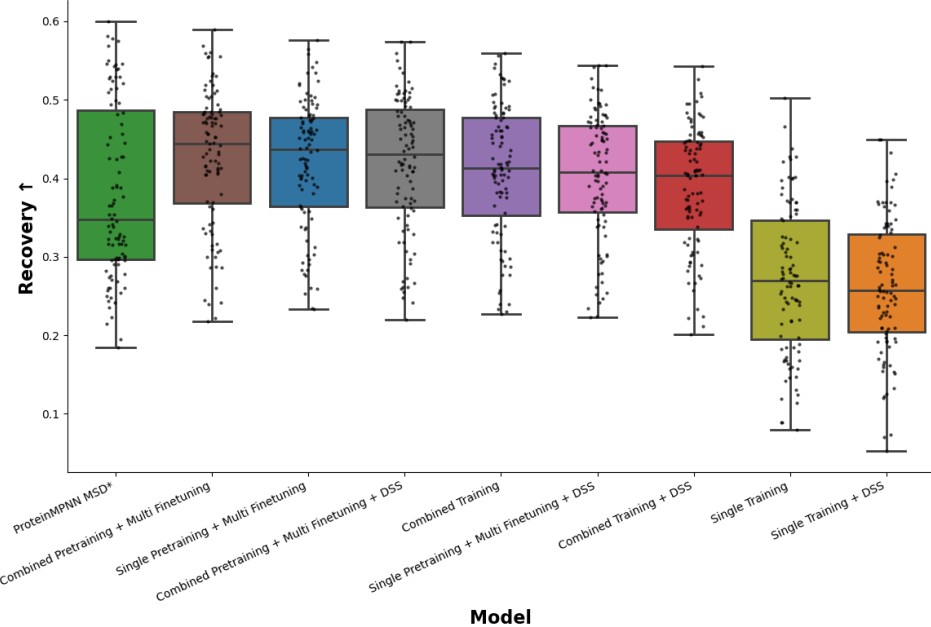

Figure 4: Sequence recovery performance across DynamicMPNN model variants and ProteinMPNN baseline on multi-state protein benchmark ($n = 96$). Combined training approaches achieve highest performance, with models that only incorporate single state training data performing poorly.

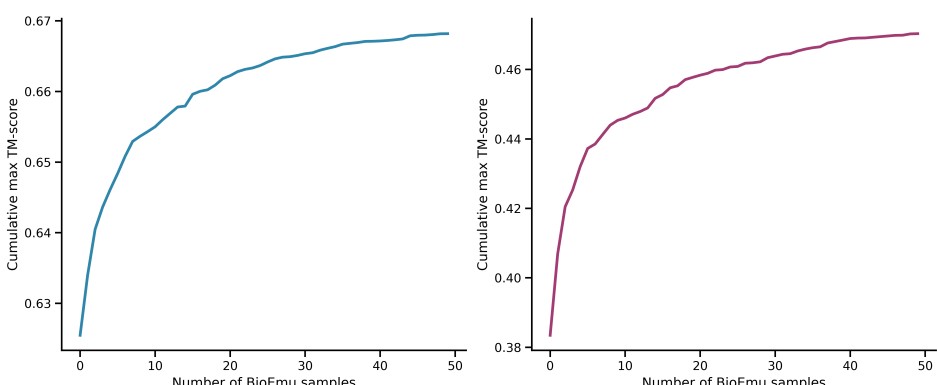

Figure 5: Convergence of cumulative maximum TM-score for BioEmu samples of Combined Training (left) and Single Training (right) model sequences. The cumulative maximum across 16 sequence designs is averaged over all test set targets.

## A.3   DETAILS ON PREVIOUS WORK

### A.3.1   COMPARISON OF MULTI-STATE INFERENCE STRATEGIES

Both ProteinMPNN-MSD (Dauparas et al., 2022; Praetorius et al., 2023) and Multi-state ESM-IF (Hsu et al., 2022) extend single-state inverse folding models to multi-state design tasks using an identical underlying mathematical strategy: step-wise logit aggregation with shared autoregressive context.

Hsu et al. (2022) formulate the objective as maximizing the geometric average of the conditional likelihoods for two states $A$ and $B$. Since sampling only depends on relative logit values (softmax

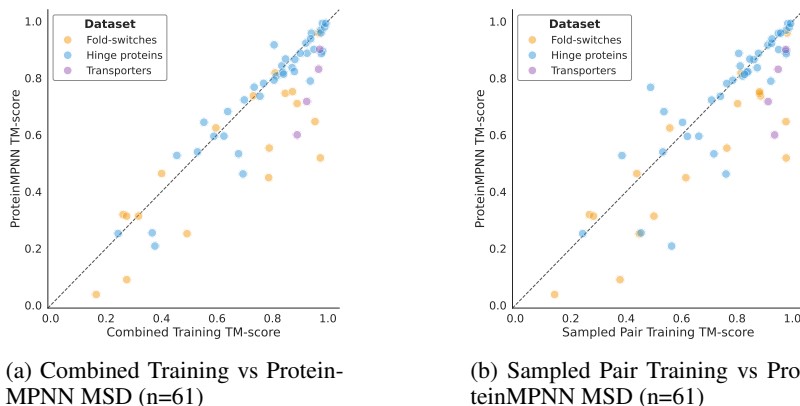

(a) Combined Training vs Protein-MPNN MSD (n=61)

(b) Sampled Pair Training vs ProteinMPNN MSD (n=61)

Figure 6: Comparison between 2 DynamicMPNN versions and ProteinMPNN MSD on protein class-stratified TM-score AF3 refoldability. The classes are Fold-switches (yellow), Hinge proteins (blue; combining the APO-HOLO, OC23, and OC85 datasets), and Transporters (purple).

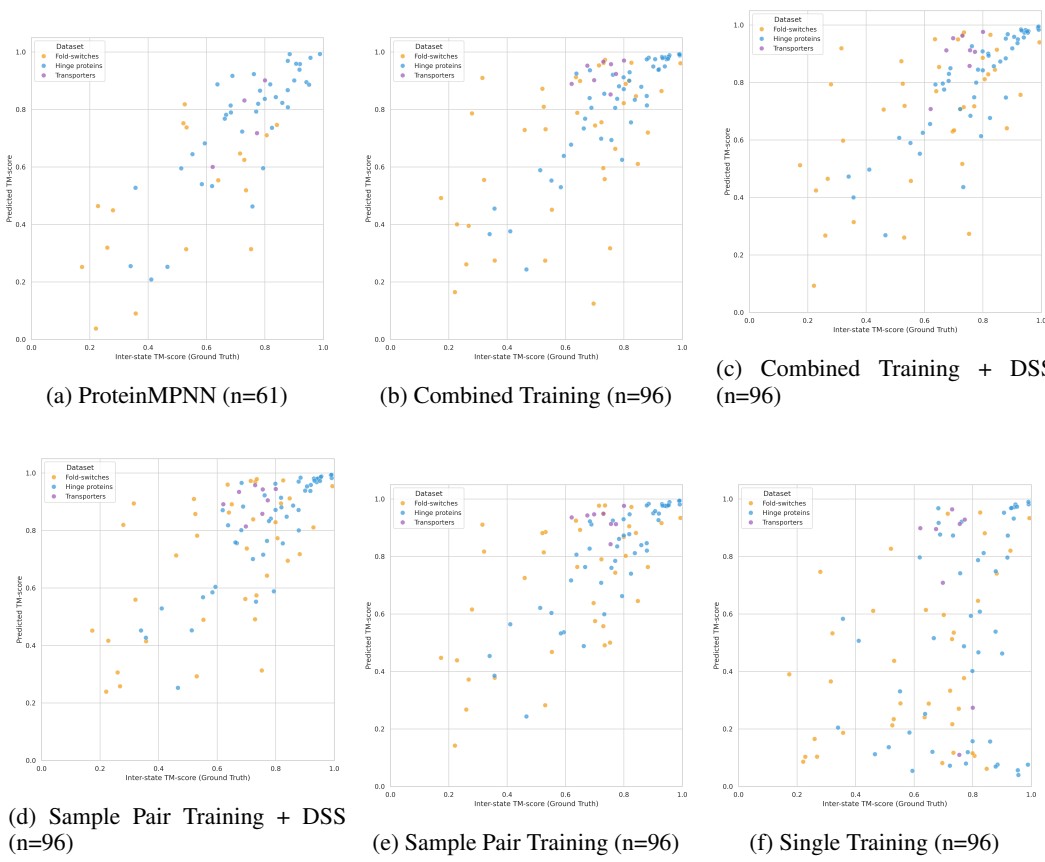

(a) ProteinMPNN (n=61)

(b) Combined Training (n=96)

(c) Combined Training + DSS (n=96)

(d) Sample Pair Training + DSS (n=96)

(e) Sample Pair Training (n=96)

(f) Single Training (n=96)

Figure 7: Distribution of Predicted template-AF3 TM-score of samples against the inter-state ground-truth TM-score. All multi-state trained model prediction accuracy correlate well with the inter-state TM-scores.

normalizes), averaging logits before sampling yields equivalent results to the geometric mean of

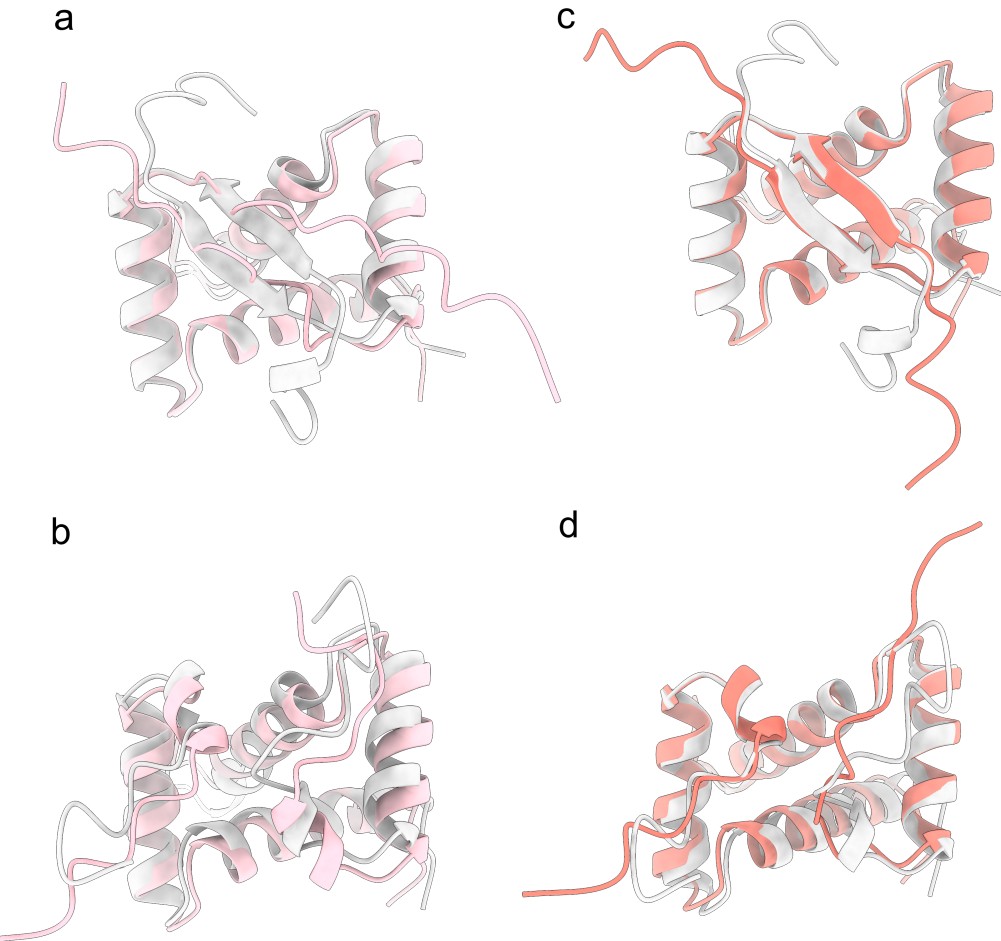

Figure 8: Switch Arc protein case study. (a, b) ProteinMPNN and (c,d) DynamicMPNN best design structure prediction (pink and salmon, respectively) against both Arc states from PDB ID: 1BDT and 1QTG respectively (grey). The DynamicMPNN design recapitulates the beta sheet fold (c), but the ProteinMPNN design0 does not (a).

probabilities:

$$\underbrace{\sqrt{P(y_t|X_A) \cdot P(y_t|X_B)}}_{\text{Geometric Mean (ESM-IF)}} \propto \exp\underbrace{\left(\frac{\text{Logits}(y_t|X_A) + \text{Logits}(y_t|X_B)}{2}\right)}_{\text{Arithmetic Mean of Logits (ProteinMPNN-MSD)}} \tag{6}$$

Consequently, both methods perform the same operation: averaging the logits from independent structural encoders at step $t$, sampling a token, and feeding that single consensus token back into the autoregressive context for all states at step $t+1$.

**Weighted aggregation.** While Hsu et al. (2022) used unweighted logit averaging, Praetorius et al. (2023) applied in their hinge protein design a 40%-60% weighting scheme to bias designs towards the effector-bound (holo) state.

### A.3.2 EXPERIMENTAL VALIDATION OF HINGE PROTEIN DESIGNS BY PRAETORIUS ET AL. (2023)

Out of the 9K designed sequences using ProteinMPNN-MSD and Rosetta (Liu & Kuhlman, 2006), 76 were randomly selected for experimental validation based on RosettaDesign (Liu & Kuhlman, 2006) and AlphaFold2 ($pLDDT > 92$, $RMSD < 1.5$, $PAE < 5$) (Jumper et al., 2021) filters.

46 out of 79 designs were expressed solubly and predominantly monomeric. Because many of the corresponding effector peptides were insoluble, only 20 out the 46 designs were tested for binding. 9 out of 20 designs showed binding via SEC experiments; advanced structural characterization with DEER and FRET confirmed the presence of the desired conformational change and peptide binding for all 8 and 3 respectively tested designs. in 3 out of these 9 designs.

## A.4 DATASET DETAILS

To construct the dataset, we obtained 46,033 Multiple Sequence Alignment (MSA) clusters at $\geq 95\%$ local sequence similarity from the latest version of CoDNaS (`v2025`) (Monzon et al., 2016), including NMR model structures. Importantly, the CoDNaS dataset prevents potential errors created by mixing different homologues in the same cluster by enforcing the same UniProt ID for all custer members. All available conformations of a protein are included - as different experimental conditions and sequence variations can reveal distinct thermodynamic states of the same protein (Best et al., 2006). While clusters contain varying numbers of conformations (Fig. 3c) we constructed our dataset using only pairs of chains from one or two PDB entries that have the largest RMSD from each cluster (Fig. 3a). Chosen pairs represent the most distinct conformational states.

While other inverse folding models (Hsu et al., 2022) saw improved performance when trained on orders of magnitude of more protein structures from AFDB, previous studies have found that the majority of high-confidence structures in AFDB map to known CATH superfamilies (Bordin et al., 2023), and that AlphaFold struggles in predicting alternative states (Chakravarty et al., 2024). We therefore decided against including AFDB structures in our training set.

PDB80 dataset, on the other hand, is meant to include more structural redundancy via its lower sequence similarity threshold (80%) at the expense of homology leakage and introduction of mutational fold-switches (He et al., 2012). These are known to adopts one state while related mutants can adopt a different state; unlike environmental switches which are the primary target of our work. We used MMseqs2 `easy-cluster` (Steinegger & Söding, 2017) to cluster all deposited SEQRES sequences in the PDB at 80% sequence identity, retaining only clusters with at least two distinct PDB entries. Clusters were filtered to exclude chains shorter than 30 or longer than 5,000 residues, as well as chains where unresolved residues (i.e. X tokens) constitute more than 40% of the sequence, yielding 46,924 multi-conformational clusters used for training via TM-score-weighted pair sampling (Section 2.1). Pairwise $C\alpha$-RMSD and TM-scores were computed between cluster members using Foldseek (Van Kempen et al., 2024).

During pair sampling in Sample Pair Training, a pair is selected with a probability as a function of the TM-score: for sequential selection of $k$ conformations, we sample each structure with probability proportional to its dissimilarity from already-selected structures:

$$P(c_i) = \frac{\exp\left(\frac{1-\overline{\mathrm{TM}}_i}{\tau}\right)}{\sum_{j\in\mathcal{C}}\exp\left(\frac{1-\overline{\mathrm{TM}}_j}{\tau}\right)} \tag{7}$$

where $\overline{\mathrm{TM}}_i = \frac{1}{|\mathcal{S}|}\sum_{s\in\mathcal{S}}\mathrm{TM}(c_i, s)$ is the average TM-score between candidate $c_i$ and the set of already-selected structures $\mathcal{S}$, $\mathcal{C}$ is the set of remaining candidates, and $\tau = 2.0$ is a temperature parameter.

### A.4.1 DATASET LIMITATIONS

Both datasets contain highly unbalanced conformational populations (Fig. 2d): a relatively small number of clusters contain thousands of sequences, while the majority contain only a few. This contributes to the relatively small intra-cluster RMSD in the majority of the clusters (Fig. 2b), since many alternative protein states live in low populations.

## A.5 MODEL DETAILS

### A.5.1 FEATURISATION SCHEME

We use a similar featurisation scheme as in (Jamasb et al., 2024). **Node scalar features** are transformer-like positional encoding in a 16-dimensional array; backbone dihedral angles

$\phi, \psi, \omega \in \mathbb{R}^6$; the virtual torsion and virtual bond angle $\kappa, \alpha \in \mathbb{R}^4$. **Node vector features** are position vectors of $C_\alpha$, $\tilde{x_i} \in \mathbb{R}^3$. **Edge scalar features** are established via k-NN (k=16) and the edge length expressed in 32 Radial Basis Functions, $e_{RBF} \in \mathbb{R}^{32}$, as well as the length of the edge itself. **Edge vector features** are edge directional unit vectors for both directions $\tilde{v_{e^{ij}}} = \tilde{x_i} - \tilde{x_j}$. To further prevent overfitting on crystallisation artifacts, random Gaussian noise ($\bar{x} = 0, \sigma = 0.1\text{Å}$) was added to the coordinates (Dauparas et al., 2022).

### A.5.2 MULTI-STATE GNN

DynamicMPNN processes one or multiple protein backbone graphs via a multi-state GNN encoder (Joshi et al., 2025). Overall, DynamicMPNN's encoder is equivariant to 3D roto-translation of coordinates as well as ordering of the states in its input. Encoding is followed by pooling node features across states, which is invariant to the ordering of the states, and autoregressive sequence decoding.

When representing conformational ensembles as a multi-graph, each node feature tensor contains three axes: (#nodes, #conformations, feature channels). Multi-state GNN's encode multi-graphs by performing message passing on the multi-graph adjacency to *independently* process each conformer, while maintaining permutation equivariance of the updated feature tensors along both the first (#nodes) and second (#conformations) axes.

### A.5.3 GEOMETRIC VECTOR PERCEPTRON LAYERS

Geometric Vector Perceptrons (GVPs) (Jing et al., 2021) are a generalization of MLPs to take tuples of scalar and vector features as input and apply $O(3)$-equivariant non-linear updates. GVP GNN layers process scalar and vector features on separate channels to maintain equivariance. The node scalars $\mathbf{s}_i \in \mathbb{R}^{k \times m}$, node vectors $\tilde{\mathbf{v}}_i \in \mathbb{R}^{k \times m' \times 3}$, and edge scalars $\mathbf{e}_{ij}$ and vectors $\tilde{\mathbf{e}}_{ij}$ communicate through a message passing operation:

$$\mathbf{m}_i, \tilde{\mathbf{m}}_i := \sum_{j \in N_i} \text{GVP}\left((\mathbf{s}_i, \tilde{\mathbf{v}}_i), (\mathbf{s}_j, \tilde{\mathbf{v}}_j), \mathbf{e}_{ij}, \tilde{\mathbf{e}}_{ij}\right), \qquad \text{(Message \& aggregate steps)} \qquad (8)$$

$$\mathbf{s}'_i, \tilde{\mathbf{v}}'_i := \text{GVP}\left((\mathbf{s}_i, \tilde{\mathbf{v}}_i), (\mathbf{m}_i, \tilde{\mathbf{m}}_i)\right). \qquad \text{(Update step)} \qquad (9)$$

The overall GNN encoder is $SO(3)$-equivariant due to the use of reflection-sensitive input features (dihedral angles) combined with $O(3)$-equivariant GVP-GNN layers.

### A.5.4 CONFORMATION ORDER-INVARIANT POOLING

After using message passing layers that are conformation order-equivariant, we add a conformation order-invariant head, which performs average pooling across the conformation channel of the scalar and vector feature tensors, similar to Joshi et al. (2025) (2025): $\mathbf{S} \in \mathbb{R}^{n \times k \times m}$ and $\tilde{\mathbf{V}} \in \mathbb{R}^{n \times k \times m' \times 3}$ to $\mathbf{S} \in \mathbb{R}^{n \times m}$ and $\tilde{\mathbf{V}} \in \mathbb{R}^{n \times m' \times 3}$, where $n$ is the sequence length, $k$ is the number of backbones, $m$ is the number of scalar features, and $m'$ is the number of vector features. The only pooling strategy used in this work is the pooling of the maximum RMSD pair of chains - therefore $k = 2$ - although more pooling strategies for homo-oligomers can be used, such as equal averaging of all chains to be inverse folded in the selected PDB entries.

### A.6 METRICS

#### A.6.1 DECOY-NORMALISED AF3 SELF-CONSISTENCY EVALUATION

**AlphaFold as a Biophysical Energy Function.** While AlphaFold (AF) excels at predicting static structures from evolutionary data, it often struggles to spontaneously sample alternative conformational states for a single sequence, effectively failing at the global *search* problem for multi-state proteins. However, recent work by Roney & Ovchinnikov (2022) established that AF has learned a robust, coevolution-independent biophysical energy function that can accurately score sequence-structure compatibility when the search space is constrained (Roney & Ovchinnikov, 2022). Specifically, they demonstrated that when a candidate structure is provided as a template, AF's output confidence metrics (pLDDT, pTM) and structural consistency (TM-score between input template

and output) correlate strongly with the actual accuracy of the model, effectively acting as a state-of-the-art energy scoring function.

Leveraging this finding, we utilize AlphaFold 3 (AF3) (Abramson et al., 2024a) not as a search engine to find the conformations, but as a scoring function to *evaluate* the compatibility of our designed sequences with the specific target geometries. By explicitly providing the target conformational state $X_k$ as a template, we direct the model to the relevant basin of the energy landscape.

**Addressing Template Bias via Decoy Normalization.** A critical challenge in template-guided evaluation is the potential for "template bias," where the model might simply copy the input geometry regardless of the sequence's actual propensity to fold into that state. To distinguish between true sequence-structure compatibility and template memorization, we introduce a **decoy-normalization** strategy.

We define a set of decoy structures $D$ that are structurally dissimilar to the target (TM-score $< 0.4$) but represent valid globular protein folds. For a designed sequence $Y$ and a target conformational state $X_k$, we calculate a normalized score that compares the structural self-consistency on the target against the self-consistency on a decoy:

$$\text{Normalized Score} = \frac{\text{Self-Consistency}(Y, X_k)}{\text{Self-Consistency}(Y, D)} \tag{10}$$

In the context of RMSD (where lower is better), this is formulated as:

$$\text{RMSD}_{\text{decoy}}(Y, X_k; D) = \frac{\text{RMSD}(\text{AF3}(Y, X_k), X_k)}{\text{RMSD}(\text{AF3}(Y, D), D)} \tag{11}$$

**Physical Interpretation.** This metric serves as a proxy for the **specificity gap** or energy gap ($\Delta E$) between the target fold and competing misfolded states.

- **High Specificity (Successful Design):** The sequence is highly compatible with the target (low RMSD / high pLDDT when prompted with $X_k$) but incompatible with the decoy (high RMSD / low pLDDT when prompted with $D$). This results in a favorable normalized score, indicating the sequence "accepts" the target fold and "rejects" the decoy.

- **Low Specificity (Hallucination/Promiscuity):** If the sequence creates a low-energy structure for both the target *and* the decoy (or high error for both), the normalized score approaches 1.0. This identifies sequences that are either generically "sticky" or for which the model is over-relying on template inputs without sequence support.

By requiring our designs to outperform decoys, we extend the "AF-as-energy-function" paradigm from simple ranking (as proposed by Roney & Ovchinnikov (Roney & Ovchinnikov, 2022)) to a rigorous **specificity filter** for *de novo* multi-state design.

