# OpenReview forum: "Multi-state Protein Sequence Design with DynamicMPNN"
_ICLR.cc/2026/Conference — ICLR 2026 Poster_

### Official Review · Reviewer_DH5T · 2025-10-19

**Soundness:** 2
**Presentation:** 2
**Contribution:** 2
**Rating:** 2
**Confidence:** 4

**Summary:**

The authors curated a dataset of similar sequences and their multiple conformations from the PDB. They trained an inverse folding model on this dataset to enable inverse folding with two target conformations. The paper proposes a new metric for evaluating multi-target inverse folding by comparing the target folded structure to the AF3-predicted structure, using the target as a template.

**Strengths:**

This paper challenges the “one sequence–one structure” assumption in inverse folding, which is an important direction to explore.
The dataset curation approach is a clever use of the existing PDB database and MSA alignments to extract sequences with multiple possible protein conformations.

**Weaknesses:**

1. Using AF3 as a validation metric to assess whether a sequence folds into the desired structure is reasonable in the single conformation case. However, in the multiple conformation setting, where AF3 is known to underperform, this approach is questionable, particularly when the target structure is provided as a template. The choice of decoy structure in such cases could significantly affect the results, and there is insufficient evidence that this metric is meaningful or correlates well with real world folding behavior.
2. The claim that the target sequence can form both target conformations is supported only by computational validation, with no experimental evidence demonstrating successful dual conformation inverse folding.
3. The paper feels somewhat rushed. The authors still have an extra page available to include additional results, while Table 2 and Figure 3 present essentially the same information, which is not an efficient use of space.

**Questions:**

1. How is the decoy structure selected for the metric? If it is chosen to be dissimilar to the targets but represents a structure the sequence is unlikely to fold into, it could significantly distort the normalized metric.

2. Why was AF3 used for evaluation, given that other models such as Boltz or AlphaFlow are known to perform better at predicting structural ensembles?


3. Line 105 broken reference

---

> ### Author Response · Authors · 2025-12-02
> **Author response (1/2)**
>
> Thank you for your feedback, which we've used to improve the manuscript. We address all your comments and concerns below.
>
> > The choice of decoy structure in such cases could significantly affect the results, and there is insufficient evidence that this metric is meaningful or correlates well with real world folding behavior.
>
> We agree that AF3 often underperforms at spontaneously sampling multiple conformations (the "search" problem). However, **our evaluation framework uses AF3 not for search, but as a biophysical energy scoring function to assess sequence-structure compatibility** - see Section 2.3 and Appendix A.6.
>
> Roney & Ovchinnikov (2022) demonstrates that when provided with a specific structure as a template, AlphaFold functions as a state-of-the-art energy function: AF's confidence metrics (pLDDT, pTM) in this template-guided mode correlate strongly with structural accuracy, effectively estimating the local free energy of that state [1].
> The reviewer correctly notes that templates introduce bias. However, this bias effectively cancels out in our normalized metric because it is applied to both the target and the decoy.
> - If AF3 purely copied templates: It would output high confidence/low RMSD for both the Target template and the Decoy template. This would result in a Normalized Score of ~1.0 (indicating a non-specific sequence).
> - Observed Behavior: We observe that for successful designs, AF3 "accepts" the Target template (high confidence) but "rejects" the Decoy template (low confidence, structural drift).
>
> > The claim that the target sequence can form both target conformations is supported only by computational validation
>
> While we agree that experimental validation is the ultimate standard, **there is strong evidence in recent literature that AlphaFold-based metrics - specifically when used with templates—serve as robust proxies for biophysical stability and experimental success:**
>  - Roney & Ovchinnikov (Phys. Rev. Lett. 2022) established that AlphaFold has learned a coevolution-independent energy function that can accurately score structural plausibility. They also demonstrated that when a candidate structure is provided as a template, AlphaFold's output metrics (pLDDT, pTM, and template-consistency) correlate strongly with the actual accuracy of that structure [1].
> - Our "normalized gain" metric operationalizes Roney & Ovchinnikov’s findings to effectively approximate the energy gap between the target state and the decoy landscape - see Section 2.3 and Appendix A.6.
> - Alignment with Established Experimental Success: Praetorius et al. (Science 2023) used AF2 metrics (pLDDT, PAE, RMSD) to filter millions of designs down to a few thousand for experimental testing [4], achieving high hit rates for hinge proteins, while Bennett et al. (Nature 2023) and Pillai et al. (Nature 2024) similarly relied on absolute AF2 thresholds to identify binders and assemblies that worked in the lab [5] [6].
>
> We have additionally run refoldability evaluations with orthogonal in silico evaluations methods (template-free AF3 and BioEmu - Table 2 and 5) and empirically demonstrate that regardless of the evaluation metric chosen, designs from models using a multi-state training strategy significantly outperform those using only single-state example.
>
> >The paper feels somewhat rushed. The authors still have an extra page available to include additional results, while Table 2 and Figure 3 present essentially the same information, which is not an efficient use of space.
>
> > Line 105 broken reference
>
> > The authors still have an extra page available to include additional results, while Table 2 and Figure 3 present essentially the same information
>
> We thank the reviewer for pointing out the broken reference and their suggestions on the manuscript's presentation. We have fixed the error, moved the redundant figure to the appendix, and made use of the full available space to include more thorough ablation studies and descriptions. We have also generally improved the polish of the manuscript.
>
> > How is the decoy structure selected for the metric?
>
> **We emphasize that our decoys are not random, implausible structures:**
> - Decoys are selected from valid protein structures that are structurally distinct from the target (TM-score < 0.4 - enforces different fold topologies) but represent physically realizable protein folds.
> - The normalized metric approximates the Energy Gap between the sequence taking the native structure and the sequence folding into a false decoy.
>
> As we mentioned in the first answer above, the bias that the false decoys introduce in the predictions correctly filters out through metric normalisation (Equation 5)  “promiscuous” sequences that agree to false templates, unrelated to the real ones. We use the bias to measure structure compatibility to good/bad templates, not to search new structures.

---

> ### Author Response · Authors · 2025-12-02
> **Author response (2/2)**
>
> > Why was AF3 used for evaluation, given that other models such as Boltz or AlphaFlow are known to perform better at predicting structural ensembles?
>
> **We have added BioEmu evaluation to the updated manuscript (Table 5), and results mirror those of our template-based AF3 metric, with Combined Training significantly outperforming Single Training.** We did not originally use BioEmu or AlphaFlow as they are restricted to single chains, precluding evaluation in the presence of binding partners that often drive conformational changes. We selected BioEmu over other ensemble generation models because it represents the current state-of-the-art, trained on orders of magnitude more data than competing methods [2, 3].
>
> Regarding Boltz, we are unaware of evidence that it improves sampling of alternate conformations; this would be surprising as it uses essentially the same architecture and training strategy as AF3.
>
>
>
> 1. Roney, J. P., & Ovchinnikov, S. (2022). State-of-the-Art Estimation of Protein Model Accuracy Using AlphaFold. Physical Review Letters, 129(23), 238101. https://doi.org/10.1103/PhysRevLett.129.238101
> 2. Lewis, Sarah, et al. "Scalable emulation of protein equilibrium ensembles with generative deep learning." Science 389.6761 (2025): eadv9817.
> 3. Jing, Bowen, Bonnie Berger, and Tommi Jaakkola. "AI-based Methods for Simulating, Sampling, and Predicting Protein Ensembles." arXiv preprint arXiv:2509.17224 (2025).
> 4. Praetorius, F., Leung, P. J. Y., Tessmer, M. H., Broerman, A., Demakis, C., Dishman, A. F., Pillai, A., Idris, A., Juergens, D., Dauparas, J., Li, X., Levine, P. M., Lamb, M., Ballard, R. K., Gerben, S. R., Nguyen, H., Kang, A., Sankaran, B., Bera, A. K., Volkman, B. F., Nivala, J., Stoll, S., & Baker, D. (2023). Design of stimulus-responsive two-state hinge proteins. Science, 381(6659), 754–760. https://doi.org/10.1126/science.adg773
> 5. Bennett, N. R., Coventry, B., Goreshnik, I., Huang, B., Allen, A., Vafeados, D., Peng, Y. P., Dauparas, J., Baek, M., Stewart, L., DiMaio, F., De Munck, S., Savvides, S. N., & Baker, D. (2023). Improving de novo Protein Binder Design with Deep Learning. Nature
> 6. Pillai, A., Idris, A., Philomin, A., Weidle, C., Skotheim, R., Leung, P. J. Y., Broerman, A., Demakis, C., Borst, A. J., Praetorius, F., & Baker, D. (2024). De novo design of allosterically switchable protein assemblies. Nature, 632(8026), 911–920. https://doi.org/10.1038/s41586-024-07813-2

---

### Official Review · Reviewer_Pioj · 2025-10-31

**Soundness:** 3
**Presentation:** 3
**Contribution:** 3
**Rating:** 6
**Confidence:** 3

**Summary:**

The paper proposes a novel approach for protein design given multiple states, which correspond to multiple structures. Instead of an ad-hoc aggregation of predictions from different states, DynamicMPNN first encodes multiple states of protein structures, along with their binding partners, into an aggregated representation in latent space. Then, an autoregressive sequence decoder is applied to decode from the pooled representation. In addition, the paper proposes a more reasonable multi-state protein design evaluation based on folding with AF3 with structures of different states as templates. The paper also provides a valuable two-state protein dataset of more than 40000 conformer pairs. Benchmarked with a wide variety of baseline models, DynamicMPNN shows superior performance in both sequence recovery and refoldability.

**Strengths:**

1.The paper presents a complete contribution to the problem of multistate protein design, covering from dataset building, model design and evaluation methods.

2.The paper is well written with the idea being clearly demonstrated.

3.The benchmark provides a thorough comparison between various combinations of models and training strategies, providing a strong support to the pretraining-finetuning training pipeline.

4.Performance on the foldability evaluation based on AF3 shows significant improvement of DynamicMPNN.

**Weaknesses:**

1.Though it is argued that refoldability with a single protein folding model prediction is not suitable for multistate design, it would still be interesting to see how the refoldability would differ if the state conformers are not provided as templates in $\mathrm{AF3}(Y, X_k)$. In other words, it would still be interesting to see the refoldability with only $\mathrm{AF3}(Y)$. If the strong bias towards a single dominant state of the existing folding models (in this case, AF3) can be demonstrated over the curated evaluation datasets, it would be valuable to support the newly proposed refoldability definition.

2.DynamicMPNN is trained and evaluated with only on two-state proteins. Extending to proteins with more states can be non-trivial since it may require more sophisticated pooling strategy across the states. DSS in this paper shows no advantage over the simple pooling strategy. The case can be different if we extend to more states, where richer interaction between conformational channels could help.

3.There are minor flaws in the presentation of the paper, see the first 2 items in the Questions section.

**Questions:**

1.The Appendix number is missing in line 105.

2.In the caption under Figure 2, the description of 2(b) and 2(c) are mismatched with the figures. Their description in the caption should be swapped.

3.How the binding partners are treated in AF3 in multi-state design evaluation.

4.Instead of using states as templates in AF3 evaluation, is it reasonable to provide only the designed sequence along with the sequence of binding partners to AF3, and use the folded structure, without using the ground truth state conformer as a template?

5.How does DynamicMPNN perform when using Single Training + MSD, using the same logit averaging strategy as ProteinMPNN-MSD? It would be interesting to see this result so that the effectiveness of unified encoding approach in multi training can be more clearly validated.

---

> ### Author Response · Authors · 2025-12-02
> **Author response**
>
> Thank you for the valuable suggestions! We appreciate that you recognise our model’s contributions to multi-state protein design as well as our performance improvements.
>
> > it would still be interesting to see how the refoldability would differ if the state conformers are not provided as templates in AF3(Y,Xk). In other words, it would still be interesting to see the refoldability with only AF3(Y).
>
> Thank you for this suggestion. **In the updated manuscript, we evaluate refoldability using template-free AF3 (Table 2)**. Combined Training outperforms Single Training across all conditions (template-free TM-score 0.78 vs 0.54; worst-state 0.69 vs 0.47), confirming that multi-state training benefits persist regardless of evaluation protocol. However, template-free AF3 shows degraded worst-state performance compared to template-based (0.69 vs 0.82), demonstrating that AF3 collapses to a single dominant conformation without structural guidance; providing strong motivation for our template-based framework.
>
> > DynamicMPNN is trained and evaluated only on two-state proteins. DSS in this paper shows no advantage over the simple pooling strategy. The case can be different if we extend to more states, where richer interaction between conformational channels could help.
>
> Thanks for the great point about model generalisability and pooling strategies. **We have comprehensively addressed model generalisability to more states** in Section 2.2.1 and 2.2.2:
> - We explain how the multi-chain model simultaneously encodes all cluster member chains in the one/two sampled PDB entries, which makes the model implicitly N-state (N>2).
> - We implement a new single-chain version of the model, with versions that explicitly encode 2, 3, or 5 states simultaneously. We analyse the benefit of adding extra states by comparing the 2-state with the 3-state, 5-state versions explicitly on designing 6 3-state or 5-state proteins (see Table 6, 7). We conclude that the optimal number of encoded states is highly dependent on the protein system to be designed.
> - We additionally test if DSS pooling offers advantages over simple pooling for the 3 and 5-state versions. Results indicate that DSS pooling provides negligible benefit over simple pooling for these tasks, suggesting that the bottleneck lies in the data availability or the protein system itself rather than the pooling expressivity.
>
> > There are minor flaws in the presentation of the paper, see the first 2 items in the Questions section.
>
> Thank you, we have corrected these issues and generally improved the quality of the presentation in the updated manuscript.
>
> > The Appendix number is missing in line 105.
>
> We fixed it, thanks!
>
> > In the caption under Figure 2, the description of 2(b) and 2(c) are mismatched with the figures. Their description in the caption should be swapped.
>
> Thank you for pointing this out, we have fixed the figure caption.
>
> > How the binding partners are treated in AF3 in multi-state design evaluation.
>
> In both the AF3-template and AF3-templateless evaluation pipeline, binding partners are included and refolded together with the designed chain.
>
>
> > is it reasonable to provide only the designed sequence along with the sequence of binding partners to AF3
>
> We have answered this issue in the first answer of this response.
>
> > How does DynamicMPNN perform when using Single Training + MSD, using the same logit averaging strategy as ProteinMPNN-MSD?
>
> The main contribution of our paper is to move away from post-hoc aggregation for multi-state protein design. Table 1 already shows that ProteinMPNN-MSD (post-hoc averaging) underperforms DynamicMPNN with Combined Training, indicating that joint multi-state learning provides benefits over post-hoc aggregation. While this specific ablation could be interesting for future work, we prioritized other experiments during the rebuttal period.

---

### Official Review · Reviewer_8aKx · 2025-11-01

**Soundness:** 4
**Presentation:** 4
**Contribution:** 3
**Rating:** 8
**Confidence:** 4

**Summary:**

This paper introduces DynamicMPNN, an inverse folding model for generating protein sequence based on multiple conformational states of a given protein. It proposes to cluster sequences and conformations in PDB to build an augmented dataset with conformational heterogeneity for training. It uses template-based AF3 for self consistency refoldability check as a metrics, and have outperformed existing inverse folding baselines.

**Strengths:**

- Develop a novel method to include explicit multi-state conditioning in the model
- Design a new data processing pipeline to effectively augment PDB for structural heterogeneity
- Rigorous evaluation: decoy normalization mitigates template bias and focuses on relative compatibility with each state.
- Clear ablations show that combined training beats single-only or multi-only; simple Deep Set pooling matches more complex DSS.

**Weaknesses:**

- Dataset uses only the max-RMSD pair, potentially biasing toward extreme transitions and discarding informative intermediates.
- Entirely in silico; mapping the normalized AF3 gains to experimental hit rates is unknown.
- Missing appendix reference (line 105)

**Questions:**

- Since there're models that predict conformational ensembles, e.g. BioEmu, AlphaFlow. Have the authors tested the generated sequences with those models, and compare with the input structures?
- While during the training, the model uses pairs of conformations from the augmented dataset. However, how much those pairs would match the distribution of practical design demands is questionable.

---

> ### Author Response · Authors · 2025-12-02
> **Author response**
>
> Thank you for your encouraging feedback! Thanks for highlighting the novelty of both the method and data, as well as rigorous evaluations and ablations.
>
> > Dataset uses only the max-RMSD pair, potentially biasing toward extreme transitions and discarding informative intermediates
>
> We have now trained 2 new DynamicMPNN versions (Pair Sample Training and Pair Sample Training + DSS) by sampling pairs of cluster members at each epoch based on a TM-score weighted probability. This way we take advantage of the full conformational information within clusters and we improve the refoldability metrics over the max-RMSD pair training - see Table 1 and 4. Check out Section 2.1 for information on the new dataset processing.
>
> > Entirely in silico; mapping the normalized AF3 gains to experimental hit rates is unknown.
>
> While we agree that experimental validation is the ultimate standard, **there is strong evidence in recent literature that AlphaFold-based metrics - specifically when used with templates—serve as robust proxies for biophysical stability and experimental success:**
> - Roney & Ovchinnikov (Phys. Rev. Lett. 2022) established that AlphaFold has learned a coevolution-independent energy function that can accurately score structural plausibility. They also demonstrated that when a candidate structure is provided as a template, AlphaFold's output metrics (pLDDT, pTM, and template-consistency) correlate strongly with the actual accuracy of that structure [1].
> - Our "normalized gain" metric operationalizes Roney & Ovchinnikov’s findings to effectively approximate the energy gap between the target state and the decoy landscape - see Section 2.3 and Appendix A.6.
> - Alignment with Established Experimental Success: Praetorius et al. (Science 2023) used AF2 metrics (pLDDT, PAE, RMSD) to filter millions of designs down to a few thousand for experimental testing [2], achieving high hit rates for hinge proteins, while Bennett et al. (Nature 2023) and Pillai et al. (Nature 2024) similarly relied on absolute AF2 thresholds to identify binders and assemblies that worked in the lab [3] [4].
>
> We have additionally run refoldability evaluations with orthogonal in silico evaluations methods (template-free AF3 and BioEmu - Table 2 and 5) and empirically demonstrate that regardless of the evaluation metric chosen, designs from models using a multi-state training strategy significantly outperform those using only single-state example.
>
> > Missing appendix reference (line 105)
>
> Fixed - thank you!
>
> > models that predict conformational ensembles, e.g. BioEmu
>
> Thanks for the suggestion, to be comprehensive, we have added BioEmu as a complementary self-consistency metric in Section 3. We see that there is strong correlation between BioEmu and our proposed AF3-template evaluation metric with the multi-state training model consistently outperforming the model trained with only single-state data across evaluation metrics.
>
>
> > how much those pairs would match the distribution of practical design demands is questionable
>
> **We have constructed a biologically meaningful and rigorous, held-out test set which does match practical design demands.** We think that as long as the model can learn useful information about protein dynamics and conformational changes, the training data is justified.
>
> This is similar to how, for training structure predictors like AlphaFold 2, the training data isolated proteins from their biological context such as ions, interaction partners. Despite this, the model learnt useful information about structure prediction of protein monomers.
>
> So overall, we think our training set design is rigorous and defensible, as long as our evaluation is sufficiently realistic (which you have agreed on in your review as well).
>
> 1. Roney, J. P., & Ovchinnikov, S. (2022). State-of-the-Art Estimation of Protein Model Accuracy Using AlphaFold. Physical Review Letters, 129(23), 238101. https://doi.org/10.1103/PhysRevLett.129.238101
> 2. Praetorius, F., Leung, P. J. Y., Tessmer, M. H., Broerman, A., Demakis, C., Dishman, A. F., Pillai, A., Idris, A., Juergens, D., Dauparas, J., Li, X., Levine, P. M., Lamb, M., Ballard, R. K., Gerben, S. R., Nguyen, H., Kang, A., Sankaran, B., Bera, A. K., Volkman, B. F., Nivala, J., Stoll, S., & Baker, D. (2023). Design of stimulus-responsive two-state hinge proteins. Science, 381(6659), 754–760. https://doi.org/10.1126/science.adg773
> 3. Bennett, N. R., Coventry, B., Goreshnik, I., Huang, B., Allen, A., Vafeados, D., Peng, Y. P., Dauparas, J., Baek, M., Stewart, L., DiMaio, F., De Munck, S., Savvides, S. N., & Baker, D. (2023). Improving de novo Protein Binder Design with Deep Learning. Nature
> 4. Pillai, A., Idris, A., Philomin, A., Weidle, C., Skotheim, R., Leung, P. J. Y., Broerman, A., Demakis, C., Borst, A. J., Praetorius, F., & Baker, D. (2024). De novo design of allosterically switchable protein assemblies. Nature, 632(8026), 911–920. https://doi.org/10.1038/s41586-024-07813-2

---

### Official Review · Reviewer_5HE4 · 2025-11-12

**Soundness:** 3
**Presentation:** 3
**Contribution:** 2
**Rating:** 4
**Confidence:** 4

**Summary:**

The paper introduces DynamicMPNN, which designs protein sequences that are compatible with multiple conformational states. Instead of aggregating single-state predictions post hoc, the model jointly learns across conformational ensembles. Using a dataset of 46,033 conformational pairs spanning 75% of CATH superfamilies and an AlphaFold–based evaluation, DynamicMPNN outperforms ProteinMPNN on a multi-state benchmark.

**Strengths:**

1. The motivation is clear: the paper tackles a well-defined gap by explicitly modeling multi-state behavior—a cornerstone of many biological functions—where post-hoc aggregation approaches have historically shown low success rates.

2. The training dataset curation is careful and comprehensive.

**Weaknesses:**

1. The empirical analysis lacks stratification by biologically and structurally meaningful factors. A breakdown by motion class (metamorphic, hinge, transporter) would contextualize the relatively high absolute RMSDs and reveal where DynamicMPNN provides the largest benefits.

2. The paper does not report scalability or runtime characteristics, leaving unclear the training and inference costs (GPU hours, memory footprint), AF3 evaluation throughput per sequence-state pair, and how computation and memory scale with the number of conformational states m.

3. The demonstrated scope is restricted to two-state systems. Although broader applicability is discussed, there is no evidence on proteins with more than two conformational states or on more continuous conformational landscapes. At least one m > 2 case study would substantiate the abstract’s claim of multi-state generality.

4. The title and text use “protein design” broadly, but the contribution is specifically an inverse folding method for protein sequence design conditioned on multiple conformations. To avoid overclaiming and to align with community terminology, the manuscript should consistently use “protein sequence design” (or “inverse folding”) where appropriate, and reserve “protein design” for pipelines that include backbone generation. The related work should be expanded to cover widely used single-state protein sequence design methods, including but not limited to ProteinMPNN, ESM-IF, CarbonDesign, and GeoEvoBuilder, with a brief comparison of how DynamicMPNN differs (e.g., multi-state conditioning, encoding of binding context, pooling across conformers) and where single-state advances might transfer or serve as baselines.

5. Writing and formatting require polish. The unresolved cross-reference “Appendix ??” (around line 105) should be fixed; the duplicated citation “Praetorius et al. (2023) (2023)” should be corrected.

**Questions:**

Please address the questions in the Weakness section.

---

> ### Author Response · Authors · 2025-12-02
> **Author response**
>
> Thank you for your very actionable feedback, which we’ve used to improve the paper. Thanks for highlighting the strong motivation and comprehensive dataset. We have addressed all your concerns below.
>
> > breakdown by motion class in results…where DynamicMPNN provides largest benefits
>
> In the updated manuscript, we’ve included this analysis in Figure 3. DynamicMPNN shows the largest gains vs. ProteinMPNN MSD for fold-switch proteins, which interestingly undergo the most complex rearrangements out of all classes. Thanks for the suggestion!
>
> > training and inference costs (GPU hours, memory footprint)
>
> We've included additional information on training (Results section) and inference costs (Appendix Table 8).
>
> > m > 2 case study would substantiate the abstract’s claim of multi-state
>
> **We have comprehensively addressed the model's capacity to handle more than two states in the revised manuscript (Sections 2.2.1, 2.2.2) and added new ablation studies (Tables 6 & 7).**
> - The architecture is inherently N-state: The DynamicMPNN architecture is permutation invariant and accepts variable input sizes. It naturally handles m > 2 states without modification.
> - Implicit vs. Explicit Training: The main multi-chain model implicitly learns from N states when a PDB entry contains multiple chains in the cluster, effectively seeing the full ensemble during training; To rigorously test scaling, we trained new single-chain versions explicitly encoding 3 and 5 states.
> - We evaluated these models on 6 proteins with complex landscapes (e.g., Calmodulin, alpha-hemolysin). We found that the optimal number of encoded states is system-dependent. For some proteins (e.g., Calmodulin), adding states improved performance; for others (e.g., MPD), it introduced noise, likely due to transient intermediate states being less informative than the stable endpoints.
> - Justification for 2-state focus: We prioritized 2-state systems for the main model due to their high biological relevance (e.g., "on/off" switches), the scarcity of high-quality >2 state training data, and the significant GPU memory costs of encoding large ensembles with binding partners.
>
>
> > manuscript should consistently use “protein sequence design”...related work should be expanded
>
> We agree, and have updated the title and manuscript. The updated related work presentation in Section 1 and in Appendix A.3 is now more comprehensive.
>
> > unresolved cross-reference…duplicate citation
>
> Thank you for pointing this out, we have fixed this and improved the general polish of the manuscript.

---

### Meta-Review · Area_Chair_5mdo · 2026-01-04

**Summary:**

DynamicMPNN replaces the "single-state-then-average" inverse-folding paradigm with end-to-end multi-conformation joint learning, explicitly encoding conformational ensembles to design protein sequences compatible with multiple structures.

**Reviewer Concerns:**

The authors’ rebuttal addressed runtime cost, multi-state experiments, BioEmu controls, and stratified analysis, yet experimental validation remains absent, decoy-selection sensitivity is unexplored, and the extrapolation of AF3 metrics lacks sufficient evidence.

**Reviewer Scores:**

Reviewers who initially gave low scores are likely to maintain their stance, as the authors added no wet-lab validation. High-scoring reviewers will probably also keep their ratings unchanged, whereas those in the middle may raise their scores.

---

### Decision · Program_Chairs · 2026-01-26

Accept (Poster)